# Brain Susceptibility to Methyl Donor Deficiency: From Fetal Programming to Aging Outcome in Rats

**DOI:** 10.3390/ijms20225692

**Published:** 2019-11-14

**Authors:** Ziad Hassan, David Coelho, Tunay Kokten, Jean-Marc Alberto, Rémy Umoret, Jean-Luc Daval, Jean-Louis Guéant, Carine Bossenmeyer-Pourié, Grégory Pourié

**Affiliations:** Faculté de médecine, INSERM 1256/University of Lorraine, F-54500 Vandoeuvre-les-Nancy, France; ziad.hassan@univ-lorraine.fr (Z.H.); david.coelho@univ-lorraine.fr (D.C.); tunay.kokten@univ-lorraine.fr (T.K.); jean-marc.alberto@univ-lorraine.fr (J.-M.A.); remy.umoret@univ-lorraine.fr (R.U.); jean-luc.daval@univ-lorraine.fr (J.-L.D.); jean-louis.gueant@univ-lorraine.fr (J.-L.G.)

**Keywords:** brain development, neuroplasticity, fetal programming, long-term effects, functional deficits

## Abstract

Deficiencies in methyl donors, folate, and vitamin B12 are known to lead to brain function defects. Fetal development is the most studied but data are also available for such an impact in elderly rats. To compare the functional consequences of nutritional deficiency in young versus adult rats, we monitored behavioral outcomes of cerebellum and hippocampus circuits in the offspring of deficient mother rats and in adult rats fed a deficient diet from 2 to 8 months-of-age. We present data showing that the main deleterious consequences are found in young ages compared to adult ones, in terms of movement coordination and learning abilities. Moreover, we obtained sex and age differences in the deleterious effects on these functions and on neuronal layer integrity in growing young rats, while deficient adults presented only slight functional alterations without tissue damage. Actually, the cerebellum and the hippocampus develop and maturate according to different time lap windows and we demonstrate that a switch to a normal diet can only rescue circuits that present a long permissive window of time, such as the cerebellum, whereas the hippocampus does not. Thus, we argue, as others have, for supplements or fortifications given over a longer time than the developmental period.

## 1. Introduction

Deficiencies in methyl donors, folate, and vitamin B12, are known to lead to defects of brain function in young ages, but they are also associated with cognitive decline, abnormal brain aging, and dementia [1,2,3]. Homocysteine, where its accumulation is a consequence of methyl donor deficiency, has also been identified as a predictor of Alzheimer′s disease [4]. The major mechanisms involve cellular effects of homocysteine, such as apoptosis, inflammation, cellular stress, and microvasculature injury [5,6,7]. Besides the immediate deleterious effects in children, such as neural tube defects and spina bifida, the biological origin of the aging effect of such a deficiency remains unclear. Moreover, recent studies pointed out putative interrelationships of multiple B vitamin deficiencies, suggesting that attention should be paid to link phenotypes to specific B vitamin deficiencies during one’s life-span [8]. The neurological effects observed in the elderly might be the consequences of deregulation of the methylation status during adult life or persistent consequences of young age modulations. In fact, deficiency in vitamin B12 and folate disturbs proper brain development by impairing neuronal growth and differentiation during young age [3,9,10], which justifies folate supplementation or fortified food for pregnant women in many countries. Such nutritional adjustments remain inefficient for inborn errors of vitamin B12 cellular metabolism in which neurocognitive impairment and psychiatric disorders appear in infants [11]. Moreover, interventional studies that aimed to evaluate the effect of folate and vitamin B12 supplements on cognitive decline in the elderly provided contradictory findings, suggesting that part of the effects could depend on the homocysteine level at baseline, which could be irreversible and/or have been produced earlier [12,13]. Indeed, early vitamin B deprivation in rat pups has been associated with a definitive atrophy of the CA1 pyramidal layer of the hippocampus, with subsequent learning and memory disabilities, even after a switch to normal food [7,14].

Considering that folate and/or B12 deficiency could produce brain defects in all ages, it would be of interest to investigate the impact of a deficient period occurring at different stages of brain development and functioning. The brain is considered to be plastic during young ages at a developmental level and more vulnerable in adult ages, especially considering the effects of strokes [15]. However, in the case of a non-traumatic pathomechanism, such as a modification of the biochemical status, fetal programming seems to be a major mechanism leading to prolonged disabilities [14,16].

To compare the long-lasting consequences on brain functioning of a diet deficient in methyl donors, we applied the same deficient diet during the developmental and suckling period in young rats and during a prolonged period of the adult life until the first stages of aging in adult rats. Such an approach could give an indication of brain vulnerability and contribute knowledge to supplementation treatments as no other successful therapeutic method is known to date.

## 2. Results

### 2.1. Gestational Deficiency Affects Young Rats During Post-Development

The linear walking test shows that pups are affected by methyl donor deficiency at the end of the suckling period (post-natal day 18 / PN18). A slight difference between males and females is recorded (Figure 1A,B). Deficient males took more time and made more errors, when running the scale, than control males (approximately three and two times more, respectively; *p* < 0.01). Concerning the females, the deficient group took a significantly longer time to reach the home-cage, but the number of errors committed appears not to be significant probably because of a larger variability in the female groups compared to the male ones.

The elevated plus maze gives an indication that male and female pups do not follow the same functional post-development. Thus, at PN 20, deficient males present a surprising behavior because they enter the maze 50% more often and for three time longer time in the open-arms of the maze compared to control males (Figure 1C–E; *p* < 0.05). Moreover, these deficient male pups moved four times less in distance when they entered the open-arms, which indicates a greater exposure to environmental risks than the controls. In addition, the number of falls from the maze are significantly higher for deficient males than for all other groups (Figure 1F; *p* < 0.01); a result that confirms the data obtained with the linear walking test (i.e., poorer coordination of locomotion in deficient males). Female pups do not present differences in this test.

The biochemical status measured in the plasma confirms the deficiency in methyl donors and the significant elevation of homocysteine at PN21 (Figure 2A; with *p* < 0.001 between deficient and control pups).

The open-field test shows that no significant differences appear between groups of pups at PN21, the weaning period. Actually, all pups, whatever the mother diet, show the same level of locomotion or exploration of the open-field and the same amount of rearing, a typical behavior of rodents in new environments (Figure 2B,C). Thus, the significant differences obtained in the test performed in the same age period would not be related to a putative difference in motivation or exploration level.

### 2.2. The Switch to a Normal Diet Restores Motor but not Cognitive Performances

At the time of weaning (PN21), all young rats were separated from their mothers and they all received a normal diet containing all vitamins. A previous study already reported that the biochemical parameters were normalized in the weeks after the switch of diet [14].

Surprisingly, the locomotion and motor coordination of 50 days-old-rats, tested on the same scale than at PN18 with the linear walking test, did not present functional defects according to the initial diet. Moreover, initially deficient rats achieved a reduction in their PN18-parameters, by 25% for the number of errors and 50% for the time to run the test, which suggests an improvement (Figure 3A,B). An exploration of the cerebellum histopathology revealed that all layers are present and that the functional outcome neurons (i.e., the Purkinje cells) show the same architecture in the molecular, body, and granular layers (Figure 3C,D) suggesting correct maturation and connection.

But despite this switch of diet, when a cognitive function, such as a learning procedure is tested, young rats from PN 32 to PN 35 showed significant differences according to the mother′s diet. Either males and females that received a deficient diet presented poorer performances than controls (Figure 3E,F). Both the escape latency to find the exit of the maze and the number of errors committed were significantly higher in initially deficient young rats than in controls of the same age from sessions 2 to 4 (from 30% to 100% increase in the escape latency and from 50% to 300% increase in errors; Figure 3E,F; *p* < 0.05). No significant differences occurred in session one because all rats were habituated but also naïve to the ideal route to exit the maze.

When memory retrieval was tested during the rat’s life-span, performances remained poor for the rats that initially received a deficient diet during their developmental and suckling periods (Figure 4A). Both males and females, initially belonging to deficient mothers, had significantly higher escape latency (almost two-times increase most of the time; *p* < 0.05) in the maze than the controls, whatever the ages tested from PN80 to PN 330 (because of the same result range, data of males and females were pooled according to the diet in order to optimize the clarity of the graph). Interestingly, initially deficient rats presented an almost three-times increase in their escape latency at day 330 (i.e., significantly higher compared to the result obtained in all other trials earlier; Figure 3A; *p* < 0.01). At the same time, control rats presented a homogenous escape latency until day 330.

In addition, the histopathological investigation of the CA1 hippocampus layer showed that both initially deficient males and females presented a reduced thickness of the CA1 layer at PN 330 (Figure 4B–D). The mature neuronal marker NeuN highlights the difference in the tissue integrity of the pyramidal layer (Figure 4E,F).

### 2.3. A Long Period of Deficiency During Adult Life Did Not Dramatically Affect Rat Behavior

The behavioral tests that revealed significant differences according to the deficient diet in young rats were used in the adult deficient model.

Surprisingly, despite of a 6-month deficient diet leading to an elevation of circulating homocysteine in both males and females (Figure 5A), deficient adult rats did not show significant differences neither in the time to run the scale, nor the number of errors committed in the linear walking test (Figure 5B,C). We only measured a significantly higher amount of errors in the linear walking test for control males compared to control females (Figure 5B; *p* < 0.05), probably due to a different strategy used to run the test. Actually, control males are those who show the quickest time to exit the scale in order to reach their home-cage.

Using the learning procedure in the aquatic maze, our results show that for cognitive performances, the long period of deficient diet in adults did not modulate the abilities of rats to efficiently run the maze. Actually, even if deficient males and females presented higher escape latencies and/or number of errors in some sessions, no significant differences were revealed (Figure 6A,B), except for deficient females in session 3, with a two-fold increase in the escape latency (Figure 6B; *p* < 0.05).

In addition, the histopathological investigation of the CA1 hippocampus layer showed that both deficient males and females present a comparable CA1 layer at the end of the 6-month period of deficiency (Figure 6C,D).

## 3. Discussion

Methyl donors deficiency leads to dramatic consequences in vulnerable periods of life, especially during development and old age [1,14,17,18]. Numerous studies focused on the development showing that gene expression is particularly affected by methyl donor deficiency, in association with abnormalities such as spina bifida and fetal death [19,20,21]. This has led to the use of supplements during a defined period of pregnancy in women in a lot of countries. However, less attention is paid to the impact of methyl donor deficiency occurring during adult life, despite that some data have shown the implication of homocysteine as a risk factor for neurodegenerative diseases [22]. Nevertheless, the central nervous system and his functions appear to be very sensitive to methyl donor deficiency. Thus, a comparison of the neuro-functional impact of such diet deficiency could highlight different putative sensitivities according to age, since brain circuitries develop and maturate in various windows of time [23].

We first submitted mother rats to a deficient diet in methyl donors, folate, and vitaminB12. Fetal development and the suckling period occurred under such deficient nutritional conditions and previous studies have shown the impact on embryos and pups in the perinatal period. Differences in gene and micro-RNA expressions are major mechanisms involved in deleterious effects, such as smaller body weight, delayed brain maturation, and the diminution of neuronal layers compared to animals fed with a normal diet [21,24,25]. Developmental pathologies could be avoided or reversed with supplementation during pregnancy. Our results presented in this study deal with two putative aspects of methyl donor deficiency on (1) the brain functions affected long after the development period [26] and (2) the differences that could occur between males and females after birth according to different time lap windows of maturation and functional acquisitions [27,28,29].

When a deficiency in methyl donors, folate, and vitamin B12 takes place during fetal development, various parameters of the late fetus and the newborn assess that the genomic program of the development are disturbed, especially for the brain. Thus, body size, weight, neural tube closure, and neuronal layers of brain sub-structures are affected [24]. Our studies and others show that brain circuitries have altered functioning in relation to neuronal plasticity [14,28] or microvasculature disturbance [7] in young ages. However, a previous study clearly showed locomotion and coordination retardation in newborn females [28], and in this present study, we show that 18-days-old males are more affected than females in these functions at the end of the suckling period (i.e., during the linear walking test and from the falls recorded in the elevated plus maze). Nevertheless, such an alteration of locomotion and coordination functions appear to be transient suggesting a disturbance of the post-natal maturation of circuits. Taken together, these data highlight sex and time differences in the functional maturation of brain circuits, which develop and maturate during the perinatal period and after birth [30].

When investigating another function, such as stress and anxiety, just before weaning on postnatal day 20, a difference between the sexes appeared. The males appeared less anxious when exploring non-protected zones than the females, representing an unusual and non-adaptative characteristic for rodents. Such an uncommon behavior represents a symptomatic trait in neurological disorders, which could be reversed by female hormonal control as already described in other topics [31,32]. These results attest that male and female neuronal circuitries maturate according to different time sequences and molecular pathways, and that the impact of methyl donor deficiency could lead to delayed functional consequences [30]. Such gender impacts within the vulnerable period of the first weeks after birth is particularly relevant to our study because we also checked the baseline activity and locomotion of males and females without recording any differences, as previously highlighted [33].

For the cognitive aspect, gestational and perinatal deficiency in methyl donors leads to a dramatic reduction of learning abilities in 35-days-old rats for either sex. Despite the switch to a normal diet at weaning, early methyl donor deficiency induces a lack of learning in a hippocampus-dependent cognitive function in both sexes. Control animals showed decreasing parameters to escape the maze, according to sessions, attesting that a learning procedure occurred. Moreover, such a functional deficit remains throughout the rat’s lifespan until the beginning of aging with an aggravation of this status at 330 days of age, in initially deficient animals. This is in relation to a lower thickness of the CA1 hippocampus layer at 330 days of age in initially deficient animals, a marker for hippocampus neuronal health [6,34].

In favor of environmental factors, adults can be exposed to prolonged methyl donor deficiencies. In contrast to the developmental period, the nervous system is fixed and the putative problems that could occur are cellular death [35] or the deregulation of functional plasticity [36].

We exposed two-month-old rats to a six-month methyl donor deficiency, but we did not reveal persistent functional deficits with behavioral approaches, neither for motor coordination nor for the learning procedure, despite effective biochemical deregulation attested by the elevation of plasma concentration of homocysteine. These results suggest that a mature brain appears more resistant than an immature one [37]. Nevertheless, some single deficits could appear during adult life but without structural changes in the brain circuits; most likely due to a modification of functional plasticity, as shown in humans [38].

Taken together, these results highlight a difference in the impacts of methyl donor deficiency according to gender and age. While huge structural and functional defects could occur at young ages, slight functional impacts could emerge in adults without strong tissue damages. Nevertheless, in adults, a disturbance of the functional plasticity could lead to an age-related decline [1,2,38]. Our study reinforces the fetal-programming theory and shows the importance to consider a larger period for interventional attempts than that of embryonic life. In the majority of cases, vitamin supplementation is stopped at birth. However, fetal-programming is a mechanism in which fetal construction of brain circuits needs environmental stimuli to improve maturation after birth. Thus, we showed that brain circuits are also vulnerable during their post-natal maturation while subjects have to acquire correct behaviors. Our results showed that the cerebellum has a wide period of several months of development, connection, and maturation (i.e., up to 90 days in rats [39]) and has a longer time lap window to reverse the deleterious effects of a deficient period than the hippocampus that has a shorter permissive period (i.e., developmental and perinatal [40]). Moreover, males and females have different age-periods of circuit connection and synapse maturation that could be prolonged until the first adult stages. In this case, a putative deficiency could lead to retardations in the acquisition of neuronal functions [28] and gives arguments in favor of a prolonged period of supplementation after the third trimester of gestation [18,25] and supplementation for an even longer time in the life-cycle [26,41].

## 4. Methods

### 4.1. Animal Treatments

Animal experiments were performed on Wistar rats (Charles River, l′Arbresle, France) and were conducted in accordance with the National Institutes of Health Guide for the Care and Use of Laboratory Animals, in an accredited establishment (Inserm U1256), according to governmental guidelines N86/609/CEE, and the local committee for ethics CELMEA according two authorized projects (october 2015: APAFIS#1498-2015082015582707v1 and march 2017: APAFIS#5509-2016053112249550v6). Adult female rats were maintained under standard laboratory conditions, on a 12-h light/dark cycle, with food and water available ad libitum. One month before pregnancy, they were fed with either standard food (*n* = 8) (Maintenance diet M20; Scientific Animal Food and Engineering, Villemoisson-sur-Orge, France) or with a diet lacking methyl donors, i.e., vitamins B12, B2, and folate (MDD, *n* = 8) (Special Diet Service, Saint-Gratien, France), described previously [14]. The amounts of the principal B vitamins are given in the table below. The assigned diet was constantly maintained until weaning of the offspring (i.e., postnatal day 21/PN21). Vitamin B6 was provided in the both diets with comparable levels. Methionine level was 0.43% in both diets and homocysteine was not detectable. All experiments were conducted on pups from mothers fed either a normal or a deficient diet.

As a comparison, adult rats were also fed the same diet (control vs. deficient) from 60 days of age to 240 days of age (*n* = 6 per group, males and females). During the last 15 days of this 6 month diet, rats were tested in the linear walking test and in the maze under the same conditions as the young rats.


**Vitamins**

**Control Food (mg/kg)**

**Deficient Food (mg/kg)**
B26.50.34B62.63.83B90.50.04B120.020.00

### 4.2. Biochemical Analyses

Plasma concentrations of vitamin B12 and folate were determined by the radio-dilution isotope assay (simulTRAC-SNB, ICN, Costa Mesa, CA, USA). Homocysteine, methylmalonic acid, and succinic acid concentrations were measured by High-Performance Liquid Chromatography (Waters, St Quentin, France) coupled to mass spectrometry (Api 4000 Qtrap, Applied Biosystems, Courtaboeuf, France) as previously described [14]. Proteins were precipitated with 0.2 N HCLO_4_, centrifuged, and the supernatant was filtered through 0.45 μm before injection into the column (Lichrospher, Merck-Millipore, Molsheim, France; with characteristics 100 RP-C18, 5 μm, 250 × 4 mm I.D.).

### 4.3. Behavioral Tests

#### 4.3.1. Linear Walking Test

This test is designed to evaluate the performances of rodents to correctly place their paws on bars (2 mm in diameter; 1 cm interval) of an horizontal scale (90 cm long). The linear walking of rats was followed by a video-tracking system (View Point, Lyon, France) to detect the body-center of the rats and the 4 paws over and under the bars. Each time a paw was detected under the bars, it was considered as an error. The time spent by each individual to run the horizontal scale was also recorded. The home-cage of each rat is placed at the end of the scale in order to increase motivation. Individuals are habituated 3 times to run the scale without any recording. The fourth run was recorded. The apparatus was cleaned with 30% alcohol between each individual.

#### 4.3.2. Elevated Plus Maze

It consists of a maze made with two corridors (9 cm large and 1 m long) crossing each other and 4 arms opposed 2 by 2. Two of these arms are open without any walls and the others two are closed by dark-grey plastic walls (35 cm high). This maze is elevated 80 cm above the floor. Individuals were placed in the middle of the crossing corridors, facing an open arm, and allowed to explore the maze for 5 min. Its behavior was recorded with a video-tracking system that allows optimum standardization. This test gives some cues reflecting the basal level of stress of individuals and information concerning the locomotion and movement coordination in the case of falls. The apparatus was cleaned with 30% alcohol between each individual.

#### 4.3.3. Open-Field Test

This test was performed to assess the amount of locomotion and the level of motivation to explore in rodents. It consists of a circular area boarded by grey-plastic walls (35 cm high). On the floor of the area, 4 quarters and 8 zones (4 near the walls and 4 in the center) were designed. Individuals were gently dropped in the middle of the open-field and followed by the video-tracking system for 5 min. The number of zones crossed and the number of rearing were recoded. The apparatus was cleaned with 30% alcohol between each individual.

### 4.4. Aquatic Maze

The maze consists of a pool filled with water (3 cm deep for young rats and 6 cm deep for adults; +/−22 °C). In this pool (1 m large, 1 m long), vertical grey-plastic walls designed 25 equal squares opened between them by doors. Closing some of these doors allows one to define an ideal route to escape the maze in at least 13 squares crossed from a starting square to an exit one. All other squares consist of dead-ends or errors. When an individual reaches the exit square, it can find its home-cage nearby heated at 30 °C. The water is used to boost motivation and to avoid cleaning between each trial in such complicated apparatus. The small level of depth is adapted to the size of the rats according to their ages and allows testing of cognitive performances rather than the ability to swim. Rats are allowed to experience the apparatus with only 1 cm of water and without any doors in order to open every square. This habituation is proposed 4 times during 2 min each and with 30 min between trials the day before the main test procedure. The main test consists of 4 trials (learning sessions) in the maze, each trial occurring everyday morning in order to respect the daily rhythm of the rats and a sufficient interval time to activate hippocampus long-term learning and memory function. Various parameters were recorded with a video-tracking system but the time to find the exit square named “escape latency” and the number of errors committed (entries in “dead-end” squares or turns-back) are the main parameters testing learning performances. Such a procedure is usually performed after weaning, in ages allowing a high level of locomotion and exploration (i.e., after PN30).

To evaluate memory retrieval, the same procedure of the learning trials can be used in punctual days during the life-span of the rats. In this memory test, the rats run the maze twice per session and the mean parameters were recorded.

### 4.5. Histology

Rats were sacrificed in different ages in order to check histopathological aspects of the brain layers of interest. Rats were anesthetized with isoflurane 4% in air flux and decapitated to proceed to brain dissection.

After 4% paraformaldehyde fixation and frozen procedure, 12 µm cryo-slides were cut in the sagittal dimension. According to the reference of the Paxinos Atlas, slides were chosen in the brain areas of interest.

Slides were counterstained with either hematoxylin or the nuclear fluorescent dye 4,6-diamidino-2-phenylindole (DAPI) (0.5 μg/mL in PBS; Sigma, F-38297 Saint-Quentin Fallavier, France). Slides were observed under fluorescence microscopy (BX51WI; Olympus, F-94593 Rungis, France) at a ×20 and ×60 magnification and pictures were performed through a digital camera and the Cell^F^ software (Soft Imaging System, Olympus).

To specifically label neurons in the different layers, tissue sections were incubated in 0.1% triton ×100 in phosphate-buffered saline (PBS) for 20 min at room temperature. Slides were dipped in PBS for 10 min, then in PBS containing 10% bovine serum for 1 h, and were incubated two days at 4 °C with primary antibodies followed by the secondary antibodies, for 1 h at room temperature (IgG conjugated to Alexa Fluor, 1/1000; Molecular Probes, Thermofisher, F-67400 Illkirch-Graffenstaden, France). Primary antibodies against NeuN for mature neurons and Calbindin-D28 for cerebellum Purkinje cells were used (1/200 dilution in PBS; Santa-Cruz Biot., Santa-Cruz, CA, USA).

### 4.6. Statistics

All data were compared using two-way ANOVA according to the sex and treatment (diet). Then, two-by-two comparisons according to the treatment were analyzed by one-way ANOVA followed by Fischer′s post-hoc analysis for *p* values. * *p* < 0.05; ** *p* < 0.01; *** *p* < 0.001.

## Figures and Tables

**Figure 1 ijms-20-05692-f001:**
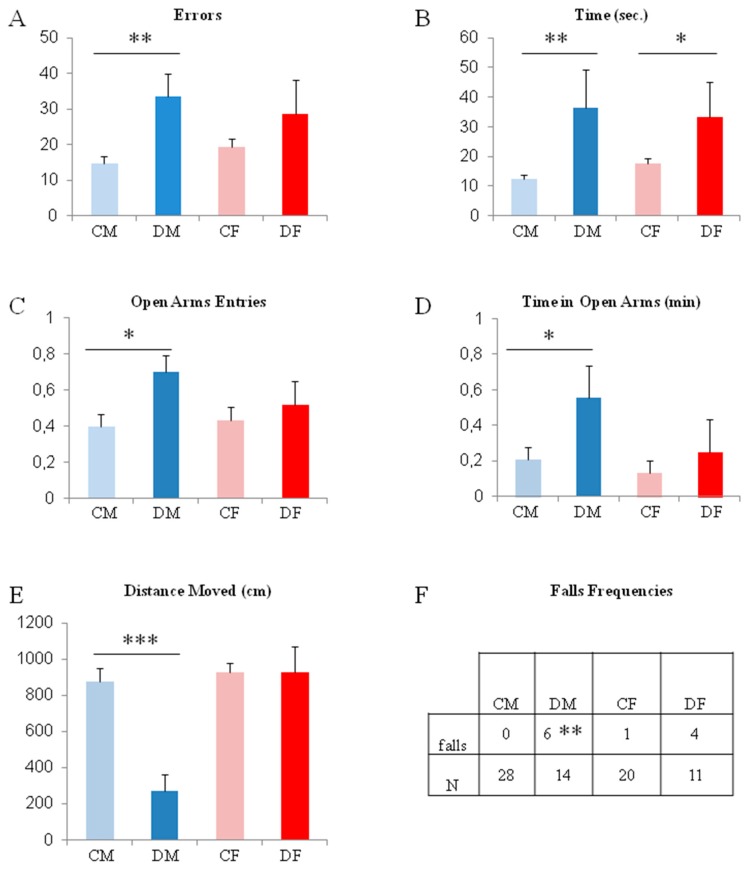
Behavioral monitoring of pups at the end of the deficient period (CM: control males; DM: deficient males; CF: control females; DF: deficient females). (**A**,**B**) Performances in the linear walking test to join the home-cage at PN 18. (**C**–**F**) Behavioral characteristics presented in the elevated plus maze at PN 20. * *p* < 0.05; ** *p* < 0.01; *** *p* < 0.001; *n* = 11 to 28 (see table **F**).

**Figure 2 ijms-20-05692-f002:**
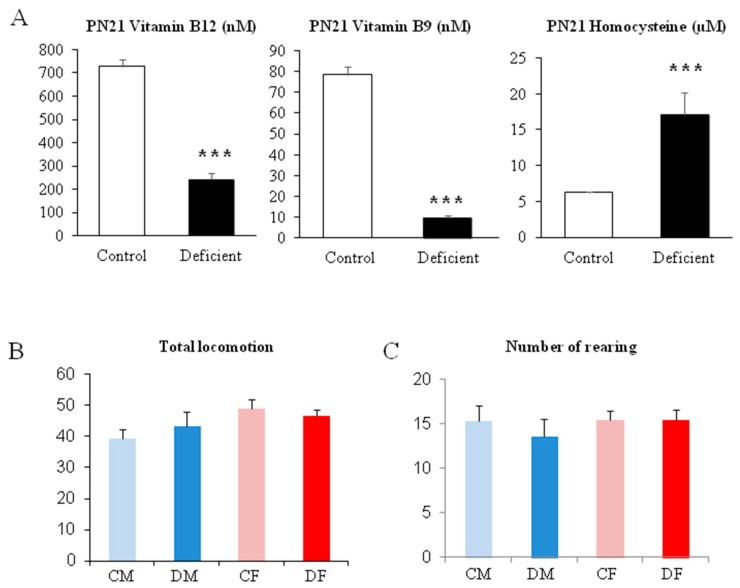
Biochemical analysis of plasma and the evaluation of the general activity at the end of the deficient period. (**A**) Plasma concentration of Vitamin B12, Vitamin B9, and homocysteine. (**B**,**C**) evaluation of locomotion and exploration levels at PN 21. *** *p* < 0.001; *n* = 11 to 28.

**Figure 3 ijms-20-05692-f003:**
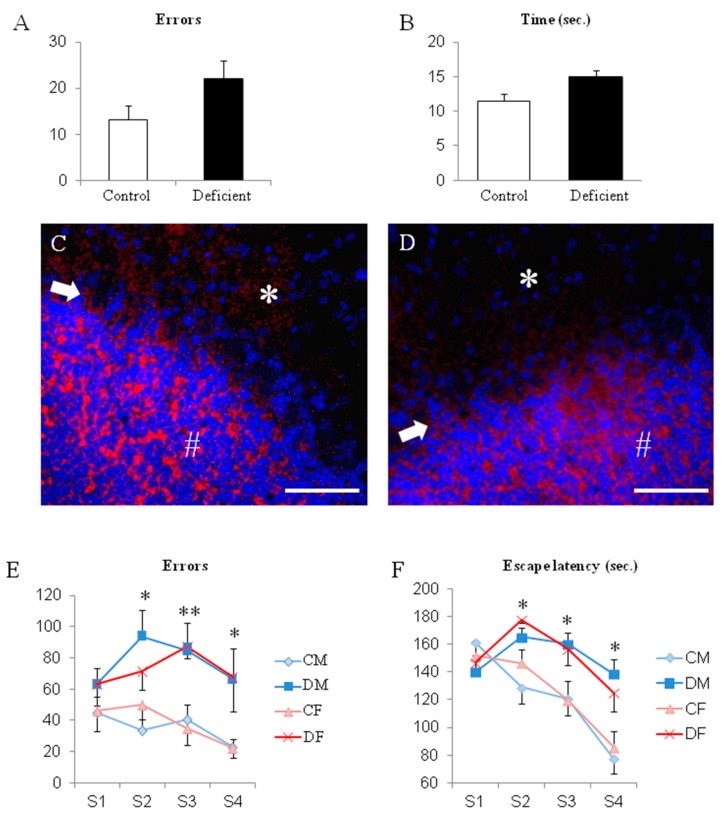
Evaluation of the motor and coordination functions after the switch to a normal diet at weaning. (**A**,**B**) Performances in the linear walking test to join the home-cage at PN 51 (no significant differences; *n* = 11 to 28). (**C**,**D**) Histopathology of the cerebellum at PN51 in control and deficient rats, respectively, with the specific marker for Purkinje cells, calbindin-D28 (red), and the nuclear dye DAPI (blue). Any differences were found whatever the layer; molecular layer (*), Purkinje bodies (arrow), granular layer/Purkinje axons (#) (×600 enlargement, bare represent 100 µm); *n* = 5 per group. (**E**,**F**) Evaluation of the learning performances in the aquatic maze at PN 32 to 35. * *p* < 0.05; ** *p* < 0.01. *n* = 8 to 11 per group.

**Figure 4 ijms-20-05692-f004:**
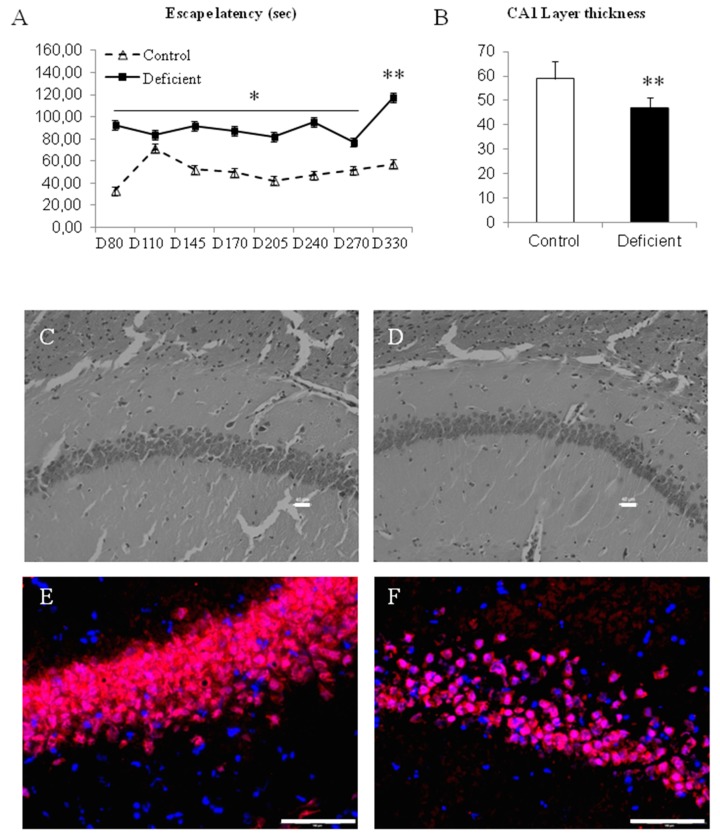
Evaluation of memory retrieval after the switch to normal diet at weaning. (**A**) Performances in the aquatic maze from PN 80 to PN 330 (escape latency in seconds with pooled results between males and females). (**B**) Measurement of the CA1 hippocampus layer thickness under a ×200 enlargement of the microscope. (**C**,**D**) Histopathology of the CA1 hippocampus layer in control and deficient rats, respectively, with hematoxyline staining (×200 enlargement, bare represent 40 µm). (**E**,**F**) Labeling with the specific marker for mature neurons NeuN (red) and the nuclear dye DAPI (blue) (×600 enlargement, bare represent 100 µm). * *p* < 0.05; ** *p* < 0.01; *n* = 8 to 11 per group.

**Figure 5 ijms-20-05692-f005:**
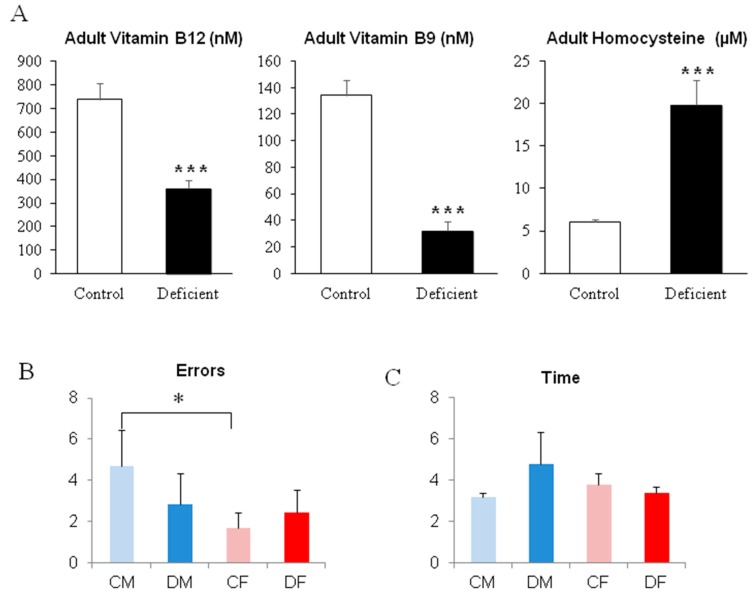
Biochemical analysis of plasma concentration of homocysteine in adults and behavioral evaluation at PN 330. (**A**) Plasma concentration of Vitamin B12, Vitamin B9, and homocysteine. (**B**,**C**) Performances in the linear walking test to join the home-cage at PN 321. * *p* < 0.05; *** *p* < 0.001; *n* = 6 per group.

**Figure 6 ijms-20-05692-f006:**
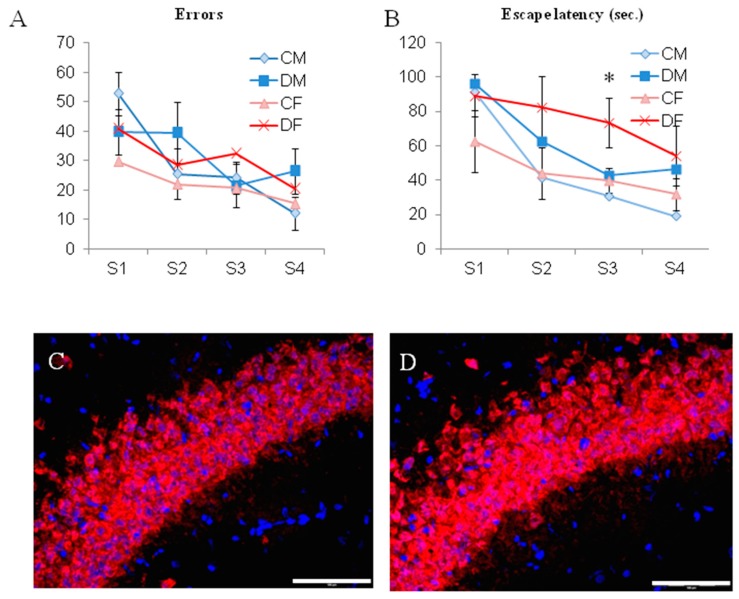
Evaluation of the learning performances in the water maze at PN 327 to 330 (**A**,**B**). (**C**,**D**) Histopathology of the CA1 hippocampus layer in control and deficient rats, respectively, with the specific marker for mature neurons NeuN (red) and the nuclear dye DAPI (blue) (×600 enlargement, bare represent 100 µm). * *p* < 0.05, *n* = 6 per group.

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
