# Peer review of "Brain Susceptibility to Methyl Donor Deficiency: From Fetal Programming to Aging Outcome in Rats"

_ijms, 2019, doi:10.3390/ijms20225692_

Round 1
Reviewer 1 Report
In my opinion, the manuscript presents interesting data about the role of folate depletion on cognitive behavior, in animals, and the results support the conclusion.
On the other, the data are not presented in an optimal way, because of lack of figure captions and numbers. Additionally, numerical values of statistical analysis are missing, in both text and figures. This aspects should be corrected.
Finally, there are some concerns regarding the description of the diet.
What does it mean normal levels in "Vitamin B6 was provided in the diet at normal levels in both groups"? In such a study, the punctual description of diet components is mandatory and should be reported in a clear way, for example in a table.
Anyway, I suggest publication with minor revisions.
Appropriate figure captions should be included. In the captions, authors should properly include the results of statistical analysis.
In the description of animal experimental procedure, authors inappropriately refer to governmental guidelines N86/609/CEE instead of EU Directive 2010/63. Additionally, the authorization number delivered by the competent institution should be included.
Author Response
Authors thank the reviewer for his suggestions.
The authorization numbers for animal experimental procedures are given page 10 line 236.
The figure captions are provided in the revised version (last page after the references) and we let the editorial board all liberty to add the figure numbers.
Now the principal numerical data are added in the "result" section at lines 65, 71 to 75, 80, 106, 112, 115, 140. They appear in relative comparison between control and deficient groups with the P-value, in order to have clear idea of the increase or decrease percentage between groups.
Page 10, the description of the diets is corrected (especially for B6) and the following table is added.
|
Vitamins |
Control food (mg/kg) |
Deficient food (mg/kg) |
|
B2 |
6.5 |
0.34 |
|
B6 |
2.6 |
3.83 |
|
B9 |
0.5 |
0.04 |
|
B12 |
0.02 |
0.00 |
Reviewer 2 Report
This research article is interesting. It indicates the importance to consider a larger period of vitamin B9 and B12 supplementation than the embryonic life.
I suggest to add in the introduction or discussion, one or two sentences on the complex relationships among B vitamins, in particular the vitamins B6, B9 and B12, in one carbon metabolism pathway. See a recent review on B vitamins by Sechi et al. in Nutrition Reviews 2016. To quote ths Review might improve the article.
The article should be extensively edited, e.g., :-Abstract, studied/studeed; page 2, line 4, "...and/or B12 DEFICIENCY (?) could…"; page 9, line 25, decreasing (?), instead of deceasing, line 38, "..mature brain appears…" etc.
Author Response
Authors agree the proposition to include the reference by Sechi et al. This reference is added in the beginning of the introduction page 1 lines 35 to 37
Authors thank the reviewer for his careful lecture and the editing corrections are made.
Reviewer 3 Report
I feel satisfied with the work done. The procedures are clear, well conducted, and the quality of the graphs and images is very good.
Author Response
Thank you very much for you kind attention.